# EXECUTION-GUIDED NEURAL PROGRAM SYNTHESIS

**Xinyun Chen**
UC Berkeley

**Chang Liu**
Citadel Securities

**Dawn Song**
UC Berkeley

## ABSTRACT

Neural program synthesis from input-output examples has attracted an increasing interest from both the machine learning and the programming language community. Most existing neural program synthesis approaches employ an encoder-decoder architecture, which uses an encoder to compute the embedding of the given input-output examples, as well as a decoder to generate the program from the embedding following a given syntax. Although such approaches achieve a reasonable performance on simple tasks such as FlashFill, on more complex tasks such as Karel, the state-of-the-art approach can only achieve an accuracy of around 77%. We observe that the main drawback of existing approaches is that the semantic information is greatly under-utilized. In this work, we propose two simple yet principled techniques to better leverage the semantic information, which are *execution-guided synthesis* and *synthesizer ensemble*. These techniques are general enough to be combined with any existing encoder-decoder-style neural program synthesizer. Applying our techniques to the Karel dataset, we can boost the accuracy from around 77% to more than 90%.

## 1 INTRODUCTION

Program synthesis is a traditional challenging problem. Such a problem typically takes a *specification* as the input, and the goal is to generate a *program* within a target *domain-specific language* (DSL). One of the most interesting forms of the specifications is input-output examples, and there have been several applications, such as FlashFill (Gulwani, 2011; Gulwani et al., 2012).

Recently, there is an increasing interest of applying neural network approaches to tackle the program synthesis problem. For example, Devlin et al. have demonstrated that using an encoder-decoder-style neural network, their neural program synthesis algorithm called RobustFill can outperform the performance of the traditional non-neural program synthesis approach by a large margin on the FlashFill task (Devlin et al., 2017b).

Despite their promising performance, we identify several inefficiencies of such encoder-decoder-style neural program synthesis approaches. In particular, such a neural network considers program synthesis as a sequence generation problem; although some recent work take the syntactical information into consideration during program generation (Bunel et al., 2018; Rabinovich et al., 2017; Yin & Neubig, 2017; Parisotto et al., 2017; Xu et al., 2017), the semantic information, which is typically well-defined in the target DSL, is not effectively leveraged by existing work.

In light of this observation, in this work, we develop simple yet principled techniques that can be combined with any existing encoder-decoder-style neural program synthesizers. The main novel technique is called *execution-guided synthesis*. The basic idea is to view the program execution as a sequence of manipulations to transform each input state into the corresponding output state. In such a view, executing a *partial program* can result in intermediate states; thus, synthesizing the rest of the program can be conditioned on these intermediate states, so that the synthesizer can take the state changes into account in the followup program generation process. Therefore, we can leverage this idea to combine with any existing encoder-decoder-style neural synthesizer, and we observe that it can significantly improve the performance of the underlying synthesizers.

In addition, we also propose a simple yet effective technique called *synthesizer ensemble*, which leverages the semantic information to ensemble multiple neural program synthesizers. In particular, for the input-output program synthesis problem, we can easily verify if a synthesized program satisfies the input-output specification, which allows us to remove invalid predictions from the ensemble

during inference time. Albeit its simplicity, to the best of our knowledge, we are not aware of any previous neural program synthesis work applying this approach. We observe that this technique further boosts the performance substantially.

We evaluate our techniques on the Karel task (Bunel et al., 2018; Devlin et al., 2017a), the largest publicly available benchmark for input-output program synthesis, on which the most performant model in the past can achieve only an accuracy of around $77\%$ (Bunel et al., 2018). We observe that our proposed techniques can gain better performance than the previous state-of-the-art results. In particular, by combining both of our techniques, we can achieve an accuracy of $92\%$, which is around 15 percentage points better than the state-of-the-art results. This shows that our approach is effective in boosting the performance of algorithms for neural program synthesis from input-output examples.

## 2  NEURAL PROGRAM SYNTHESIS FROM INPUT-OUPUT EXAMPLES

In this section, we first introduce the input-output program synthesis problem and existing encoder-decoder-style neural program synthesis approaches, then present an overview of our approaches.

### 2.1  PROBLEM DEFINITION

We follow the literature (Devlin et al., 2017b; Bunel et al., 2018; Chen et al., 2018) to formally define the input-output program synthesis problem below.

**Problem Definition 1 (Program emulation)** *Let $\mathcal{L}$ be the space of all valid programs in the domain-specific language (DSL). Given a set of input-output pairs $\{(I^k, O^k)\}_{k=1}^K$ (or $\{IO^K\}$ in short), where there exists a program $P \in \mathcal{L}$, such that $P(I^k) = O^k, \forall k \in \{1, ..., K\}$. Our goal is to compute the output $O^{\text{test}}$ for a new test input $I^{\text{test}}$, so that $O^{\text{test}} = P(I^{\text{test}})$.*

Although the problem definition only requires to compute the output for a test input, a typical method is to synthesize a program $P' \in \mathcal{L}$ such that $P'$ is consistent with all input-output pairs $\{IO^K\}$, and then use $P'$ to compute the output. In this case, we say *program $P'$ emulates the program $P$ corresponding to $\{IO^K\}$*.

In particular, in this work, we are mainly interested in the following formulation of the problem.

**Problem Definition 2 (Program synthesis)** *Let $\mathcal{L}$ be the space of all valid programs in the domain-specific language (DSL). Given a training dataset of $\{IO^K\}_i$ for $i = 1, ..., N$, where $N$ is the size of the training data, compute a* synthesizer *$\Gamma$, so that given a test input-output example set $\{IO^K\}_{\text{test}}$, the synthesizer $\Gamma(\{IO^K\}_{\text{test}}) = P$ produces a program $P$, which emulates the program corresponding to $\{IO^K\}_{\text{test}}$.*

### 2.2  ENCODER-DECODER-STYLE NEURAL PROGRAM SYNTHESIS APPROACHES

There have been many approaches proposed for different neural program synthesis tasks, and most of them follow an encoder-decoder-style neural network architecture (Bunel et al., 2018; Devlin et al., 2017b; Parisotto et al., 2017). Figure 1 shows a general neural network architecture for input-output program synthesis. First, an encoder converts input-output examples $\{IO^K\}$ into an embedding. For example, RobustFill (Devlin et al., 2017b) deals with the string transformation tasks, thus it uses LSTMs as the encoder. For the Karel task, both inputs and outputs are 2D grids (see Figure 2); therefore, Bunel et al. (2018) employ a CNN as the encoder.

Once the IO embeddings are computed, these approaches employ an LSTM decoder to generate the programs conditioned on the embeddings. For program synthesis, one unique property is that the generated program should satisfy the synthax of $\mathcal{L}$. Therefore, a commonly used refinement is to filter syntactically invalid program prefixes during generation (Parisotto et al., 2017; Devlin et al., 2017b; Bunel et al., 2018).

In the above approaches, only the syntax information is leveraged; the semantics of $\mathcal{L}$ is not utilized. In particular, standard supervised training procedure could suffer from *program aliasing*: for the same input-output examples, there are multiple semantically equivalent programs, but all except the one provided in the training data will be penalized as wrong programs. To mitigate this issue, Bunel et

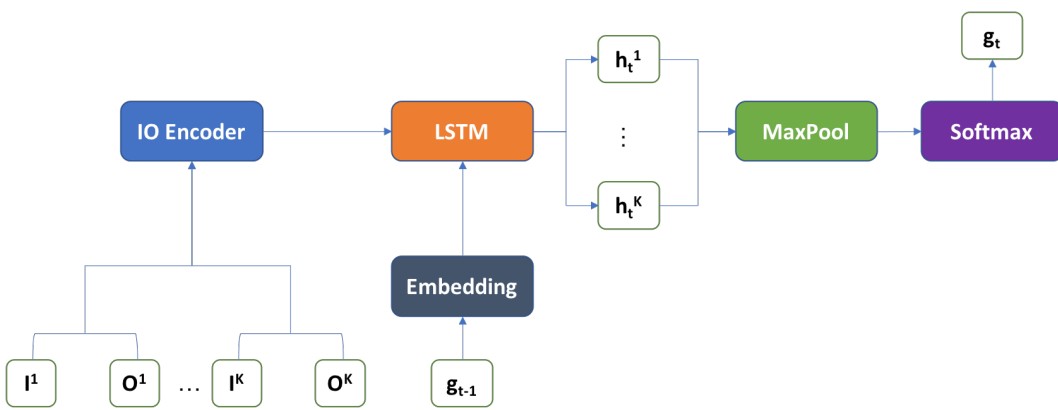

Figure 1: A neural network architecture for input-output program synthesis (e.g., (Bunel et al., 2018)). At each timestep $t$, the decoder LSTM generates a program token $g_t$ conditioned on both the input-output pairs $\{IO^K\}$ and the previous program token $g_{t-1}$. Each IO pair is fed into the LSTM individually, and a max-pooling operation is performed over the hidden states $\{h_t^k\}_{k=1}^K$ of the last layer of LSTM for all IO pairs. The resulted vector is fed into a softmax layer to obtain a prediction probability distribution over all the possible program tokens in the vocabulary. More details can be found in Appendix C.

**Program:** turnLeft() ; move() ; putMarker()

**States:**

Figure 2: An example of the execution of partial programs to reach the target state in the Karel domain. The blue dot denotes the marker put by the Karel robot.

al. propose to train with reinforcement learning, so that it rewards all semantically correct programs once they are fully generated (Bunel et al., 2018). In our work, we demonstrate that we can leverage the semantic information in an effective way that provides a better performance.

## 2.3 AN OVERVIEW OF OUR APPROACHES

In this work, we propose two general and principled techniques that can improve the performance over existing work, which are *execution-guided synthesis* (Section 3) and *synthesizer ensemble* (Section 4). The main idea of our techniques is to better leverage the semantics of the language $\mathcal{L}$ during synthesis. Meanwhile, our techniques are compatible with any existing encoder-decoder-style neural program synthesis architecture. We will describe these techniques in detail in the following sections.

## 3 EXECUTION-GUIDED SYNTHESIS

Existing approaches generate the entire program only based on the input-output examples before execution. However, this is an inefficient use of the semantics of $\mathcal{L}$. For example, when a program consists of a sequence of statements, we can view the output to be a result by continuously executing each statement in the sequence to convert the input state into a sequence of intermediate states. Figure 2 illustrates such an example. From this perspective, instead of generating the whole program at once, we can generate one statement at a time based on the intermediate/output state pairs.

However, most interesting programs are not just sequential. In this work, we explore this idea using a general control-flow framework. In particular, given any language $\mathcal{L}$, we extend it with three classical types of control-flow: sequential, branching, and looping. The extended language is call $\mathcal{L}_{ext}$. Then, we develop our above idea based on $\mathcal{L}_{ext}$, called *execution-guided synthesis*. In the following, to make our discussion concise, we first formalize $\mathcal{L}_{ext}$ (Section 3.1), and then present the idea of execution-guided synthesis (Section 3.2).

$$\text{S-Stmt}\frac{\langle \mathcal{S}, s\rangle \Downarrow s'}{\langle \mathcal{S}, s\rangle \rightarrow \langle \bot, s'\rangle} \qquad \text{S-Seq}\frac{\langle B_1, s\rangle \rightarrow \langle B_1', s'\rangle}{\langle B_1; B_2, s\rangle \rightarrow \langle B_1'; B_2, s'\rangle}$$

$$\text{S-Seq-Bot } \langle \bot; B_2, s\rangle \rightarrow \langle B_2, s\rangle \qquad \text{S-If}\frac{\langle \mathcal{C}, s\rangle \Downarrow b \quad i = \begin{cases} 1 & b \text{ is true} \\ 2 & b \text{ is false} \end{cases}}{\langle \textbf{if } \mathcal{C} \textbf{ then } B_1 \textbf{ else } B_2 \textbf{ fi}, s\rangle \rightarrow \langle B_i, s\rangle}$$

$$\text{S-While } \langle \textbf{while } \mathcal{C} \textbf{ do } B \textbf{ end}, s\rangle \rightarrow \langle \textbf{if } C \textbf{ then } B; \textbf{while } C \textbf{ do } B \textbf{ end else } \bot \textbf{ fi}, s\rangle$$

Table 2: Semantic rules $\langle B, s\rangle \rightarrow \langle B', s'\rangle$ for $\mathcal{L}_{\text{ext}}$.

### 3.1 THE FORMAL SPECIFICATION OF THE EXTENDED LANGUAGE $\mathcal{L}_{\text{ext}}$

In this work, we assume some additional control-flow syntax on top of $\mathcal{L}$. We define the extended language $\mathcal{L}_{\text{ext}}$ in Table 1. In particular, we assume that a code block $B$ can be composed by a sequence of statements $\mathcal{S} \in \mathcal{L}$ or sub code blocks, and each code block can also be an if-statement or a while-statement. We use $\mathcal{C}$ to indicate a condition expression in $\mathcal{L}$, and $\bot$ to indicate the termination of a program execution.

$$
\begin{aligned}
P \quad &:= \quad B; \bot \\
B \quad &:= \quad \bot \mid \mathcal{S} \mid B; B \\
&\quad \mid \quad \textbf{if } \mathcal{C} \textbf{ then } B \textbf{ else } B \textbf{ fi} \\
&\quad \mid \quad \textbf{while } \mathcal{C} \textbf{ do } B \textbf{ end} \\
\mathcal{S}, \mathcal{C} \quad &\in \quad \mathcal{L}
\end{aligned}
$$

Table 1: Syntax of $\mathcal{L}_{\text{ext}}$.

The semantics of $\mathcal{L}_{\text{ext}}$ is specified in Table 2. These rules are largely standard following the convention in programming language literature. In particular, each rule's name starts with S- indicating that they are semantics rules; the suffixes indicate the constructors each rule specifies, (e.g., Stmt for statements, Seq for sequences, etc.). The judgment $\langle B, s\rangle \rightarrow \langle B', s'\rangle$ indicates a small-step execution of program $B$ over state $s$ to result in a new program $B'$ and a new state $s'$. The judgments $\langle \mathcal{S}, s\rangle \Downarrow s'$ and $\langle C, s\rangle \Downarrow b$ capture the big-step execution in $\mathcal{L}$ that statement $\mathcal{S}$ evaluates to a new state $s'$ from $s$, and condition $\mathcal{C}$ evaluates to a boolean value $b$ from $s$. Following the semantics of $\mathcal{L}_{\text{ext}}$, we can formally define a program execution.

**Definition 1 (Program execution)** *Given a program $P \in \mathcal{L}_{\text{ext}}$ and an input $I$, the execution is a sequence $s_0...s_T$, such that (1) $s_0 = I$; (2) $B_0 = P$; (3) $\langle B_i, s_i\rangle \rightarrow \langle B_{i+1}, s_{i+1}\rangle$ for $i = 0, ..., T-1$; and (4) $B_T = \bot$. The output of the program is $O = s_T$.*

### 3.2 EXECUTION-GUIDED SYNTHESIS ALGORITHM

In Definition 1, we can observe that the initial and final states are simply two special states provided as the input-output examples of the synthesis problem. Thus, a synthesizer $\Gamma$ for input-output pairs should also be able to take any state-pairs as inputs. Our execution-guided synthesis algorithm takes advantage of this fact to improve the performance of the synthesizer. In the following, we discuss three cases from the easiest to the hardest to present our approach.

**Sequential programs.** We now consider the simplest case, where the program is in the form of $\mathcal{S}_1; ...; \mathcal{S}_T$, to illustrate the basic idea of execution-guided synthesis algorithm. We present the algorithm in Algorithm 1. Assuming the input-output examples are $\{IO^K\}$, we can treat them as $K$ state-pairs $\{(s_{\textbf{i}}^k, s_{\textbf{o}}^k)\}_{k=1}^K$, where $s_{\textbf{i}}^k = I^k, s_{\textbf{o}}^k = O^k$. The Exec algorithm takes the synthesizer $\Gamma$ and the input-output pairs $\{IO^K\}$ as its input. It also takes an additional input $\Delta$, which is the *ending token* to be synthesized. For the top-level program, $\Delta$ will be the $\bot$ token. Later we will see that when synthesizing the sub-program for If- and While-blocks, different ending tokens will be used.

The synthesized program is initially empty (line 3). Then the algorithm iteratively generates one statement $\mathcal{S}$ at a time (line 4-10 and 17), and appends it to the end of $P$ (line 16). Importantly, if $\mathcal{S}$ is not an if-statement or a while-statement, for which we handle separately, the algorithm executes the newly generated statement $\mathcal{S}$ to transit $s_{\textbf{i}}^k$ into $s_{\textbf{new}}^k$ (line 11-14). Therefore, in the subsequent iteration, the synthesizer can start from the new states $s_{\textbf{new}}^k$ after executing the partial program $P$

---

**Algorithm 1** Execution-guided synthesis (sequential case)

---

1: **function** EXEC($\Gamma, \{(s_{\mathbf{i}}^k, s_{\mathbf{o}}^k)\}_{k=1}^K, \Delta$)
2:     // The main algorithm is called using Exec ($\Gamma, \{IO^K\}, \bot$)
3:     $P \leftarrow \bot$
4:     $\mathcal{S} \leftarrow \Gamma(\{(s_{\mathbf{i}}^k, s_{\mathbf{o}}^k)\}_{k=1}^K)$
5:     **while** $S \neq \Delta$ **do**
6:         **if** $\mathcal{S} = $ **if**-token **then**         // If-statement synthesis
7:             $\mathcal{S}, \{(s_{\mathbf{i}}^k, s_{\mathbf{o}}^k)\}_{k=1}^K \leftarrow \text{ExecIf}(\Gamma, \{(s_{\mathbf{i}}^k, s_{\mathbf{o}}^k)\}_{k=1}^K)$
8:         **else**
9:             **if** $\mathcal{S} = $ **while**-token **then**         // While-statement synthesis
10:                 $\mathcal{S}, \{(s_{\mathbf{i}}^k, s_{\mathbf{o}}^k)\}_{k=1}^K \leftarrow \text{ExecWhile}(\Gamma, \{(s_{\mathbf{i}}^k, s_{\mathbf{o}}^k)\}_{k=1}^K)$
11:             **else**     // Execution of $\mathcal{S}$
12:                 $\langle \mathcal{S}, s_{\mathbf{i}}^k \rangle \rightarrow \langle \bot, s_{\mathbf{new}}^k \rangle$ for $k = 1, ..., K$
13:                 $s_{\mathbf{i}}^k \leftarrow s_{\mathbf{new}}^k$ for $k = 1, ..., K$
14:             **end if**
15:         **end if**
16:         $P \leftarrow P; \mathcal{S}$
17:         $\mathcal{S} \leftarrow \Gamma(\{(s_{\mathbf{i}}^k, s_{\mathbf{o}}^k)\}_{k=1}^K)$
18:     **end while**
19:     **return** $P$
20: **end function**

---

**Algorithm 2** Execution-guided synthesis (if-statement)

---

1: **function** EXECIF($\Gamma, \mathcal{I}$)
2:     $\mathcal{C} \leftarrow \Gamma(\mathcal{I})$
3:     $\mathcal{I}_{\mathbf{t}} \leftarrow \{(s_{\mathbf{i}}, s_{\mathbf{o}}) \in \mathcal{I} | \langle \mathcal{C}, s_{\mathbf{i}} \rangle \Downarrow \text{true}\}$
4:     $\mathcal{I}_{\mathbf{f}} \leftarrow \{(s_{\mathbf{i}}, s_{\mathbf{o}}) \in \mathcal{I} | \langle \mathcal{C}, s_{\mathbf{i}} \rangle \Downarrow \text{false}\}$
5:     $B_{\mathbf{t}} \leftarrow \text{Exec}(\Gamma, \mathcal{I}_{\mathbf{t}}, \textbf{else-token})$
6:     $B_{\mathbf{f}} \leftarrow \text{Exec}(\Gamma, \mathcal{I}_{\mathbf{f}}, \textbf{fi-token})$
7:     $\mathcal{I}_{\mathbf{t}}' \leftarrow \{(s_{\mathbf{new}}, s_{\mathbf{o}}) | (s_{\mathbf{i}}, s_{\mathbf{o}}) \in \mathcal{I}_{\mathbf{t}} \wedge \langle B_{\mathbf{t}}, s_{\mathbf{i}} \rangle \Downarrow s_{\mathbf{new}}\}$
8:     $\mathcal{I}_{\mathbf{f}}' \leftarrow \{(s_{\mathbf{new}}, s_{\mathbf{o}}) | (s_{\mathbf{i}}, s_{\mathbf{o}}) \in \mathcal{I}_{\mathbf{f}} \wedge \langle B_{\mathbf{f}}, s_{\mathbf{i}} \rangle \Downarrow s_{\mathbf{new}}\}$
9:     $\mathcal{I} \leftarrow \mathcal{I}_{\mathbf{t}}' \cup \mathcal{I}_{\mathbf{f}}'$
10:     $\mathcal{S} \leftarrow$ **if** $C$ **then** $B_{\mathbf{t}}$ **else** $B_{\mathbf{f}}$ **fi**
11:     **return** $\mathcal{S}, \mathcal{I}$
12: **end function**

---

generated so far. In doing so, the synthesizer can see all intermediate states to better adjust the followup synthesis strategies to improve the overall synthesis performance.

**Branching programs.** Dealing with if-statements is slightly more complicated than sequential programs, since in an if-statement, not all statements will be executed on all inputs. Following the execution in Algorithm 1 naively, we have to use $\Gamma$ to synthesize the entire if-statement before being able to execute the partially generated program to derive intermediate states.

Therefore, in Algorithm 2, we extend the above idea to handle if-statements. When the next predicted token is an **if**-token, our execution-guided synthesizer first predicts the condition of the if-statement $\mathcal{C}$ (line 2). Then, we evaluate $\mathcal{C}$ over all state-pairs. Based on the evaluation results, we can divide the IO pairs into two sets $\mathcal{I}_{\mathbf{t}}$ and $\mathcal{I}_{\mathbf{f}}$ (line 3-4), so that all states in the former meet the branching condition to go to the true branch, and all states in the latter go to the false branch. Therefore, in the followup synthesis, we do not need to consider $\mathcal{I}_{\mathbf{f}}$ (or $\mathcal{I}_{\mathbf{t}}$) when synthesizing the true branch (or the false branch) (line 5-6). Note that in line 5-6, synthesizing both true-branch and false-branch employ execution-guided synthesis algorithm to leverage intermediate states, and different ending tokens (i.e., **else** and **fi**) are supplied respectively. Once we have done the synthesis of both branches, we can execute the generated branches to get the new states $\mathcal{I}$ (line 7-9), and return the newly generated if-statement and updated states to the caller of this algorithm.

In Algorithm 2, we use $\langle B, s \rangle \Downarrow s'$ to indicate a big-step execution of code block $B$ over state $s$ to get $s'$. In particular, this means that $\langle B, s \rangle \rightarrow \langle B_1, s_1 \rangle \rightarrow ... \rightarrow \langle \bot, s' \rangle$.

**Looping programs.** The remaining problem is to handle while-statements. Due to the rule S-While (see Table 2), a while-statement

$$\textbf{while } \mathcal{C} \textbf{ do } B \textbf{ end} \tag{1}$$

is equivalent to

$$\textbf{if } \mathcal{C} \textbf{ then } (B; \textbf{while } \mathcal{C} \textbf{ do } B \textbf{ end}) \textbf{ else } \bot \textbf{ fi} \tag{2}$$

Therefore, we can employ a procedure similar to Algorithm 2 once a **while**-token is predicted. However, there are two differences. First, in (2), the false-branch is empty, thus we do not need to deal with the false-branch. Second, although the true-branch is $B; \textbf{while } \mathcal{C} \textbf{ do } B \textbf{ end}$, once we have generated $B$, we do not need to generate the rest of the true-branch, since both $\mathcal{C}$ and $B$ have been generated. The detailed algorithm can be found in Appendix B.

**Remarks.** The final algorithm is called by Exec $(\Gamma, \{IO^K\}, \bot)$. Note that our execution-guided synthesis algorithm can be applied to any neural synthesizer $\Gamma$, and we can train the synthesizer $\Gamma$ using any supervised or reinforcement learning algorithms that have been proposed before (Devlin et al., 2017b; Bunel et al., 2018). In our evaluation, we demonstrate that our execution-guided synthesis technique helps boost the performance of both supervised and reinforcement learning algorithms proposed in existing work (Bunel et al., 2018).

## 4 Synthesizer Ensemble

In our experiments, we observe that when we use different random initializations of the synthesizer during training, even if the synthesizer architectures are the same, they will be effective on different subsets of the dataset, although the overall prediction accuracy is similar to each other. Thus, a natural idea is to train multiple synthesizers, and ensemble them to build a more performant synthesizer.

Different from other deep learning tasks, for program synthesis task, without knowing the ground truth, we can already filter out those wrong predictions that cannot satisfy the input-output specification. Thus, we ensemble multiple synthesizers as follows: we run all synthesizers to obtain multiple programs, and select from programs that are consistent with all input-output examples. This provides us with a better chance to select the correct final prediction that generalizes to held-out IO pairs.

The main subtlety of such an approach is to deal with the case when multiple generated programs satisfy the input-output examples. In this work, we consider several alternatives as follows:

- **Majority vote.** We can choose the most frequently predicted program as the final prediction.
- **Shortest.** Following the Occam's razor principle, we can choose the shortest program as the final prediction.

## 5 Evaluation

In this section, we demonstrate the effectiveness of our approaches on the Karel dataset (Pattis, 1981; Bunel et al., 2018). We first introduce the task, discuss the experimental details, and present the results.

### 5.1 The Karel Task

Karel is an educational programming language proposed in the 1980s (Pattis, 1981). Using this language, we can control a robot to move inside a 2D grid world and modify the world state, and our goal is to synthesize a program given a small number of input and output grids as the specification. Such tasks have been used in Stanford CS introductory courses (CS106A, 2018) and the Hour of Code (HoC, 2018), and have been studied recently in several neural program synthesis works (Devlin et al., 2017a; Bunel et al., 2018; Shin et al., 2018). Figure 2 shows an example in the Karel domain. We provide the grammar specification and the state representation in Appendix A. In particular, the

| Training | | Ensemble | Generalization | Exact Match | From |
|---|---|---|---|---|---|
| MLE | SL | - | 71.91% | 39.94% | Bunel et al. (2018) |
| | | S | 78.80% | 46.68% | This work |
| | | MV | 78.80% | **47.08%** | This work |
| | RL | - | 77.12% | 32.17% | Bunel et al. (2018) |
| | | S | 84.84% | 46.04% | This work |
| | | MV | 84.16% | 46.36% | This work |
| Exec | SL | - | 85.08% | 40.88% | This work |
| | | S | 91.60% | 45.84% | |
| | | MV | 91.52% | 45.36% | |
| | RL | - | 86.04% | 39.40% | |
| | | S | 91.68% | 46.36% | |
| | | MV | **92.00%** | 45.64% | |

Table 3: Accuracy on the Karel test set. In the "Training" column, we use "MLE" and "Exec" to indicate the training approaches proposed in (Bunel et al., 2018) and this work, "SL" and "RL" to indicate supervised learning and reinforcement learning respectively. In the "Ensemble" column, dash indicates that no ensemble is used, "S" and "MV" indicate the shortest and majority vote principles respectively. For the single model accuracy, we report the results of the model with the best generalization accuracy. We include 15 models in each ensemble.

Karel DSL includes control flow constructs such as conditionals and loops, which is more challenging than problems well-studied before, such as FlashFill (Gulwani, 2011; Devlin et al., 2017b).

Our evaluation follows the setup in (Bunel et al., 2018). We train and evaluate our approaches on their dataset, which is built by randomly sampling programs from the DSL. For each program, 5 IO pairs serve as the specification, and the sixth one is the held-out test sample. In total, there are 1,116,854 programs for training, 2,500 in the validation set, and 2,500 in the test set. We evaluate the following two metrics, which are the same as in (Bunel et al., 2018):

- **Exact Match.** The predicted program is an *exact match* if it is the same as the ground truth.
- **Generalization.** The predicted program is a *generalization* if it satisfies the input-output examples in both the specification and the held-out examples.

**Training dataset construction for the Exec algorithm.** Note that the original Karel dataset only provides the input-output examples and the ground truth programs. To train the synthesizer $\Gamma$ with our Exec algorithm in the supervised learning setting, we need the supervision on intermediate states as well, which can be obtained easily by executing the ground truth programs following the semantics (Table 2). In particular, for each sample $\langle \{IO^K\}, P \rangle$ in the original training set, we construct a sequence of training samples $\langle \{(s_{i-1}^k, O^k)\}_{k=1}^{K_i}, S_i \rangle$ $(i = 1, 2, ..., T)$, with each sample containing $K_i \leq K$ state pairs and a program $S_i \in P$. The algorithm to construct this set is largely analogous to the semantics specification, and we defer the details to Appendix B.

**Training algorithms.** Once the training set is constructed, the neural synthesizer $\Gamma$ can be trained on this new dataset using the same algorithm as the one for training $\Gamma$ on the original dataset. Therefore, our Exec algorithm can be applied to both supervised learning and reinforcement learning algorithms proposed in (Bunel et al., 2018) for evaluation.

**Model details.** We employ the same neural network architecture as in (Bunel et al., 2018) to synthesize the programs, which is briefly discussed in Section 2.2. During the inference time, we set the beam size $B = 64$, and select the one with the highest prediction probability from the remaining programs. More details can be found in Appendix C.

## 5.2 RESULTS

We present our main results in Table 3. We report the results of ensembling 15 models for our ensemble techniques. For reference, we include MLE and RL results in (Bunel et al., 2018), which were the state-of-the-art on the Karel task for the exact match and generalization metrics respectively.

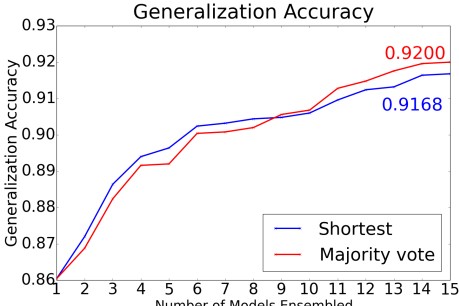 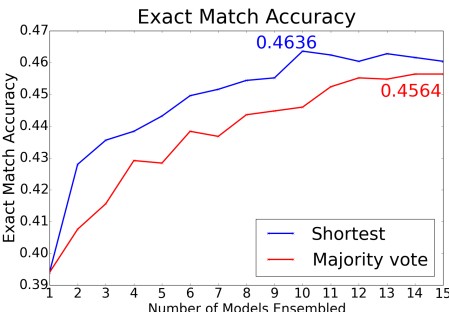

Figure 3: Results of the ensemble model trained with Exec + RL approach. Left: generalization accuracy. Right: exact match accuracy. The corresponding figures using models trained with Exec approach can be found in Appendix D.2.

We first apply our ensemble techniques to these approaches, and observe that the performance could be significantly boosted by around 7%.

We next observe that our execution-guided synthesis alone can significantly improve the generalization accuracy over all approaches from (Bunel et al., 2018), even after we accompany their approaches with our ensemble techniques. In particular, without the ensemble, "Exec+SL" already improves "MLE+RL" by 8 points on generalization accuracy; and when ensemble approaches are applied to "MLE+RL", this gap is shrunk, but still positive. Similar to (Bunel et al., 2018), we can also train our Exec model using the RL technique, which improves the generalization accuracy by another 1 point, while slightly decreases the exact match accuracy. These results show that utilizing intermediate execution states alone is already an effective approach to boost the performance.

Note that the improvement of Exec on the exact match accuracy is relatively minor, and sometimes negative when applying the ensemble to baseline training algorithms. This is because our Exec algorithm is not designed to optimize for exact match accuracy. In particular, we decouple the full programs in the original training dataset into small pieces, thus our synthesizer $\Gamma$ is trained with segments of the original training programs instead of the full programs. In doing so, our synthesizer is more capable of generating programs piece-by-piece and thus tends to generate semantically correct programs (i.e., with a better generalization accuracy) rather than the same programs in the training and testing sets (i.e., with a better exact match accuracy). In fact, for real-world applications, the generalization accuracy is more important than the exact match accuracy, because exact match accuracy is more about evaluating how well the synthesizer recovers the language model of the pCFG sampler used to generate the dataset. More discussion can be found in Appendix D.1.

Finally, we apply our ensemble approaches on top of Exec+SL and Exec+RL. We observe that this can further improve the generalization accuracy and exact match accuracy by around 6% on top of the best single model. These results show that our ensemble approaches consistently boost the performance, regardless of the underlying models used for ensembling.

In addition, we investigate the performance of ensembling different number of models. We present the results of ensembling Exec + RL models in Figure 3, and defer the results of ensembling Exec models to Appendix D.2. We observe that using the shortest principle is generally more effective than using the majority vote principle, especially when fewer number of models are included in the ensemble. However, when there are more models, majority vote may achieve a better generalization accuracy than the shortest principle. This is reasonable, since when there are too few models, there might not be enough effective models to form the majority.

Interestingly, we observe that Exec+RL+Ensemble does not significantly improve the performance over Exec+SL+Ensemble. This may be due to that the improvement from ensemble hides the improvement from RL. More discussion of our evaluation results can be found in Appendix D.

To summarize, we make the following key observations:

1. Our execution-guided synthesis technique can effectively improve previous approaches, which only use the syntactic information, or the final program execution outputs.

2. Our ensemble approaches can effectively improve the performance regardless of the underlying models being used.

3. The different modules of our proposed approaches, i.e., execution-guided synthesis and ensemble techniques, can work independently to improve the performance, and thus they can be applied independently to other tasks as well.

4. By combining all our novel techniques, we improve the state-of-the-art on the Karel task by 14.88 points (generalization) and 7.14 points (exact match).

## 6 RELATED WORK

Synthesizing a program from input-output examples is an important challenging problem with many applications (Devlin et al., 2017b; Gulwani et al., 2012; Gulwani, 2011; Bunel et al., 2018; Chen et al., 2018; Cai et al., 2017; Li et al., 2017; Reed & De Freitas, 2016; Zaremba et al., 2016; Zaremba & Sutskever, 2015; Fox et al., 2018; Xiao et al., 2018; Ganin et al., 2018). There has been an emerging interest in studying neural networks for program synthesis. A line of work studies training a neural network to directly generate the outputs given the inputs (Devlin et al., 2017b;a; Graves et al., 2014; Joulin & Mikolov, 2015; Kaiser & Sutskever, 2015). In particular, Devlin et al. study the Karel domain (Devlin et al., 2017a). However, as shown in (Devlin et al., 2017a), this approach is incapable of handling the case when the number of input-output examples is small, and is hard to generalize to unseen inputs.

Recent work study using neural networks to generate programs in a domain-specific language (DSL) from a few input-output examples (Devlin et al., 2017b; Bunel et al., 2018; Parisotto et al., 2017; Polosukhin & Skidanov, 2018; Zohar & Wolf, 2018). Several work synthesize programs for FlashFill tasks, which are in the string transformation domain (Devlin et al., 2017b; Parisotto et al., 2017). Other work synthesize programs in a LISP-style DSL for array manipulation (Polosukhin & Skidanov, 2018; Zohar & Wolf, 2018). In particular, Zohar & Wolf (2018) also study the idea of encoding the state of the transformed inputs as it is updated during execution. However, these DSLs only include sequential programs, and do not support more complex control flows such as loops and conditionals in our studied Karel problem. Prior works also consider incorporating syntax constraints and information from program execution to facilitate program synthesis (Devlin et al., 2017b; Wang et al., 2018a; Bunel et al., 2018). However, all these works generate the whole program, and use its execution results to guide the synthesis process; in contrast, our work leverages more fine-grained yet generic semantic information that can be gathered during executing programs in most imperative languages. As a result, our approach's performance is significantly better than previous work (Bunel et al., 2018).

Previous work also study program synthesis given intermediate states. For example, Sun et al. (2018) propose to synthesize the program from demonstration videos, which can be viewed as sequences of states. In such a problem, all intermediate states can be extracted from the videos. On the contrary, in the input-output program synthesis problem studied in our work, the input to the program synthesizer provides only the initial state and the final state. Thus, our synthesizer is required to address the challenge of inferring intermediate states, which is mainly tackled by our execution-guided synthesis algorithm.

In contrast to training a neural network to generate the entire program, a recent line of research studies using a neural network to guide the symbolic program search based on the input-output specification, so that the search process prioritizes the operators that have higher domain-specific scores predicted by the neural networks (Balog et al., 2017; Vijayakumar et al., 2018). Instead of predicting such domain-specific scores to guide the program search, we directly incorporate the domain knowledge by executing partial programs, and utilize the execution results for program generation of the neural network synthesizer. Meanwhile, recent work propose to leverage the full execution traces in the context of program repair (Wang et al., 2018b). Our work is the first to leverage partial execution traces for the program synthesis task, which is a much harder task than program repair.

# 7   CONCLUSION

In this work, we propose two general and principled techniques to better leverage the semantic information for neural program synthesis: (1) execution-guided synthesis; and (2) synthesizer ensemble. On a rich DSL with complex control flows, we achieve a significant performance gain over the existing work, which demonstrates that utilizing the semantic information is crucial in boosting the performance of neural program synthesis approaches. We believe that our techniques are also beneficial to other program generation applications, and we consider extending our techniques to handle programming languages with richer semantics as important future work. At the same time, we have observed that utilizing existing reinforcement learning techniques does not provide much performance gain when combined with our approaches. We believe that there is plenty of room left for further improvement, and we are also interested in exploring this problem in the future.

## ACKNOWLEDGEMENT

We thank the anonymous reviewers for their valuable comments. This material is in part based upon work supported by the National Science Foundation under Grant No. TWC-1409915, Berkeley DeepDrive, and DARPA D3M under Grant No. FA8750-17-2-0091. Any opinions, findings, and conclusions or recommendations expressed in this material are those of the author(s) and do not necessarily reflect the views of the National Science Foundation.

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

## A    MORE DESCRIPTIONS OF THE KAREL DOMAIN

Figure 4 presents the grammar specification of the Karel DSL.

Each Karel grid world has a maximum size of $18 \times 18$, and is represented as a $16 \times 18 \times 18$ tensor, where each cell of the grid is represented as a 16-dimensional vector corresponding to the features described in Table 4.

```
Prog p   ::=   def run() : s
Stmt s   ::=   while(b) : s | repeat(r) : s | s₁ ; s₂ | a
         |     if(b) : s | ifelse(b) : s₁ else : s₂
Cond b   ::=   frontIsClear() | leftIsClear() | rightIsClear
         |     markersPresent() | noMarkersPresent() | not b
Action a ::=   move() | turnRight() | turnLeft()
         |     pickMarker() | putMarker()
Cste r   ::=   0 | 1 | ... | 19
```

Figure 4: Grammar for the Karel task.

| |
|---|
| Robot facing North |
| Robot facing East |
| Robot facing South |
| Robot facing West |
| Obstacle |
| Grid boundary |
| 1 marker |
| 2 markers |
| 3 markers |
| 4 markers |
| 5 markers |
| 6 markers |
| 7 markers |
| 8 markers |
| 9 markers |
| 10 markers |

Table 4: Representation of each cell in the Karel state.

## B    MORE DETAILS ABOUT THE EXECUTION-GUIDED ALGORITHM

**While-statement synthesis algorithm.**    Algorithm 3 demonstrates the execution-guided algorithm for while-statement synthesis.

**Training dataset construction for supervised learning.**    Consider a program $P = S_1; ...; S_T; \bot$, where each $S_i$ is in one of the following forms: (1) $S_i \in \mathcal{L}$; (2) **if** $\mathcal{C}$ **then** $B_\mathbf{t}$ **else** $B_\mathbf{f}$ **fi**; and (3) **while** $\mathcal{C}$ **do** $B$ **end**. For each $S_i \in \mathcal{L}$, we construct a sample of $\langle \{(s_{i-1}^k, O^k)\}_{k=1}^K, S_i \rangle$ directly.

For $S_i =$ **if** $\mathcal{C}$ **then** $B_\mathbf{t}$ **else** $B_\mathbf{f}$ **fi**, we first construct a training sample $\langle \{(s_{i-1}^k, O^k)\}_{k=1}^K,$ **if** $\mathcal{C}$ **then**$\rangle$. Afterwards, we split the input-output examples into two subsets $\mathcal{I}_\mathbf{t} \uplus \mathcal{I}_\mathbf{f}$, where for all $(s, O) \in \mathcal{I}_\mathbf{t}$, we have $\langle \mathcal{C}, s \rangle \Downarrow$ true; and $\mathcal{C}$ evaluates to false for all $(s, O) \in \mathcal{I}_\mathbf{f}$ on the other hand. Then we obtain two derived samples $\langle \mathcal{I}_\mathbf{t}, B_\mathbf{t}$ **else**; $S_{i+1}; ...; S_T; \bot \rangle$ and $\langle \mathcal{I}_\mathbf{f}, B_\mathbf{f}$ **fi**; $S_{i+1}; ...; S_T; \bot \rangle$, from which we construct training samples respectively using the same approach as discussed above.

In a similar way, we can deal with $S_i =$ **while** $\mathcal{C}$ **do** $B$ **end**. Finally, we include $\langle \{(O^k, O^k)\}_{k=1}^K, \bot \rangle$ in our constructed training set.

---

**Algorithm 3** Execution-guided synthesis (while-statement)

1: **function** ExecWhile($\Gamma, \mathcal{I}$)
2:     $\mathcal{C} \leftarrow \Gamma(\mathcal{I})$
3:     $\mathcal{I}_t \leftarrow \{(s_i, s_o) \in \mathcal{I} | \langle \mathcal{C}, s_i \rangle \Downarrow \text{true}\}$
4:     $\mathcal{I}_f \leftarrow \{(s_i, s_o) \in \mathcal{I} | \langle \mathcal{C}, s_i \rangle \Downarrow \text{false}\}$
5:     $B_t \leftarrow \text{Exec}(\Gamma, \mathcal{I}_t, \textbf{end}\text{-token})$
6:     $\mathcal{I}'_t \leftarrow \{(s_{new}, s_o) | (s_i, s_o) \in \mathcal{I}_t \wedge \langle B_t, s_i \rangle \Downarrow s_{new}\}$
7:     $\mathcal{I} \leftarrow \mathcal{I}'_t \cup \mathcal{I}_f$
8:     $\mathcal{S} \leftarrow \textbf{while } \mathcal{C} \textbf{ do } B_t \textbf{ end}$
9:     **return** $\mathcal{S}, \mathcal{I}$
10: **end function**

---

## C    Model Details

### C.1    Neural Network Architecture

Our neural network architecture can be found in Figure 1, which follows the design in Bunel et al. (2018). In particular, the IO Encoder is a convolutional neural network to encode the input and output grids, which outputs a 512-dimensional vector for each input-output pair. The decoder is a 2-layer LSTM with a hidden size of 256. The embedding size of the program tokens is 256.

Each program is represented as a sequence of tokens $G = [g_1, g_2, ..., g_L]$, where each program token $g_i$ belongs to a vocabulary $\Sigma$. At each timestep $t$, the decoder LSTM generates a program token $g_t$ conditioned on both the input-output pair and the previous program token $g_{t-1}$, thus the input dimension is 768. Each IO pair is fed into the LSTM individually, and we a max-pooling operation is performed over the hidden states $\{h_t\}_{k=1}^K$ of the last layer of LSTM for all IO pairs. The resulted 256-dimensional vector is fed into a softmax layer to obtain a prediction probability distribution over all the 52 possible program tokens in the vocabulary.

Notice that this neural network architecture can also be applied to other program synthesis problems, with modifications of the IO encoder architectures for different formats of input-output pairs. For example, in the domain where input-output examples are text strings, such as FlashFill (Gulwani, 2011), the IO encoders can be recurrent neural networks (RNNs) (Devlin et al., 2017b).

### C.2    Training Objective Functions

To estimate the parameters $\theta$ of the neural network, we first perform supervised learning to maximize the conditional log-likelihood of the referenced programs (Parisotto et al., 2017; Devlin et al., 2017b; Bunel et al., 2018). In particular, we estimate $\theta^*$ such that

$$\theta^* = \arg\max_\theta \prod_{i=1}^N p_\theta(\pi_i | \{IO_i^k\}_{k=1}^K) = \arg\max_\theta \sum_{i=1}^N \log p_\theta(\pi_i | \{IO_i^k\}_{k=1}^K) \tag{3}$$

Where $\pi_i$ are the ground truth programs provided in the training set.

When training with reinforcement learning, we leverage the policy gradient algorithm REIN-FORCE (Williams, 1992) to solve the following objective:

$$\theta^* = \arg\max_\theta \sum_{i=1}^N \sum_G \log p_\theta(G | \{IO_i^k\}_{k=1}^K) R_i(G) \tag{4}$$

Where $R_i(G)$ is the reward function to represent the quality of the sampled program $G$. In our evaluation, we set $R_i(G) = 1$ if $G$ gives the correct outputs for given inputs, and $R_i(G) = 0$ otherwise.

**Ground truth:**    putMarker(); while (markersPresent()): move()

**Prediction:**    putMarker(); move()

**States:**

$s_0^1 (I^1)$          $s_1^1$          $s_2^1 (O^1)$

$s_0^2 (I^2)$          $s_1^2$          $s_2^2 (O^2)$

Figure 5: An example of the predicted program that generalizes to all input-output examples, but is different from the ground truth. Here, we only include 2 out of 5 input-output examples for simplicity. Notice that the predicted program is simpler than the ground truth.

### C.3 TRAINING HYPER-PARAMETERS

We use the Adam optimizer (Kingma & Ba, 2015) for both the supervised training and the RL training. The learning rate of supervised training is $10^{-4}$, and the learning rate of reinforcement learning is $10^{-5}$. We set the batch size to be $128$ for supervised training, and $16$ for RL training.

## D EVALUATION DETAILS

### D.1 MORE ANALYSIS OF EVALUATION RESULTS

In our evaluation, we observe that while our Exec approach significantly boosts the generalization accuracy, the performance gain of the exact match accuracy is much smaller, and sometimes even negative. We attribute this to the fact that the ground truth program in the Karel benchmark is not always the simplest one satisfying the input-output examples; on the other hand, our approach tends to provide short predictions among all programs consistent with the input-output specification. For example, Figure 5 shows a predicted program that is simpler than the ground truth, while also satisfies the input-output pairs. Notice that different from the MLE approach in (Bunel et al., 2018), our model is not directly optimized for the exact match accuracy, since the training set not only includes the input-output examples in the original training set, but also the intermediate state pairs, which constitute a larger part of our augmented training set. Meanwhile, in our training set, for state pairs resembling $\{(s_1^k, s_2^k)\}_{k=1}^K$ in Figure 5, it is more common for the ground truth program to be a single "move()" statement than other more complicated ones. Therefore, when training with our approach, the model is less prone to overfitting to the sample distribution of the original dataset, and focuses more on the program semantics captured by the intermediate execution.

### D.2 MORE DETAILS OF THE ENSEMBLE

For different training approaches of a single model, we train 15 models with different random initializations. To do the ensemble, we first sort the 15 models according to the descending order of their generalization accuracies on the validation set, then select the first $k$ models to compute the results of the $k$-model ensemble. When multiple programs satisfy the ensemble criterion, e.g., with the shortest length for the *Shortest* method, we choose the one from the models with better generalization accuracies on the validation set.

Figure 6 shows the results of the ensemble trained with our Exec approach. Tables 5 and 6 show the numerical results of applying ensemble to our Exec approach and Exec + RL approach.

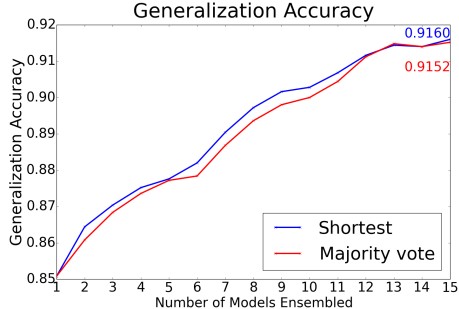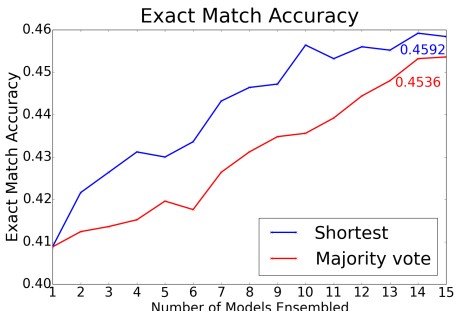

Figure 6: Results of the ensemble model trained with our Exec approach. Left: generalization accuracy. Right: exact match accuracy.

| Ensemble | 1 | 2 | 3 | 4 | 5 | 6 | 7 | 8 | 9 | 10 | 11 | 12 | 13 | 14 | 15 |
|---|---|---|---|---|---|---|---|---|---|---|---|---|---|---|---|
| Exec (S) | 40.88% | 42.16% | 42.64% | 43.12% | 43.00% | 43.36% | 44.32% | 44.64% | 44.72% | 45.64% | 45.32% | 45.60% | 45.52% | 45.92% | 45.84% |
| Exec (MV) | 40.88% | 41.24% | 41.36% | 41.52% | 41.96% | 41.76% | 42.64% | 43.12% | 43.48% | 43.56% | 43.92% | 44.44% | 44.80% | 45.32% | 45.36% |
| Exec + RL (S) | 39.40% | 42.80% | 43.56% | 43.84% | 44.32% | 44.96% | 45.16% | 45.44% | 45.52% | 46.36% | 46.24% | 46.04% | 46.28% | 46.16% | 46.04% |
| Exec + RL (MV) | 39.40% | 40.76% | 41.56% | 42.92% | 42.84% | 43.84% | 43.68% | 44.36% | 44.48% | 44.60% | 45.24% | 45.52% | 45.48% | 45.64% | 45.64% |

Table 5: Exact match accuracy of the ensemble.

| Ensemble | 1 | 2 | 3 | 4 | 5 | 6 | 7 | 8 | 9 | 10 | 11 | 12 | 13 | 14 | 15 |
|---|---|---|---|---|---|---|---|---|---|---|---|---|---|---|---|
| Exec (S) | 85.08% | 86.44% | 87.04% | 87.52% | 87.76% | 88.20% | 89.04% | 89.72% | 90.16% | 90.28% | 90.68% | 91.16% | 91.44% | 91.40% | 91.60% |
| Exec (MV) | 85.08% | 86.08% | 86.84% | 87.36% | 87.72% | 87.84% | 88.68% | 89.36% | 89.80% | 90.00% | 90.44% | 91.12% | 91.48% | 91.40% | 91.52% |
| Exec + RL (S) | 86.04% | 87.20% | 88.64% | 89.40% | 89.64% | 90.24% | 90.32% | 90.44% | 90.48% | 90.60% | 90.96% | 91.24% | 91.32% | 91.64% | 91.68% |
| Exec + RL (MV) | 86.04% | 86.88% | 88.24% | 89.16% | 89.20% | 90.04% | 90.08% | 90.20% | 90.56% | 90.68% | 91.28% | 91.48% | 91.76% | 91.96% | 92.00% |

Table 6: Generalization accuracy of the ensemble.

