# OpenReview forum: "Execution-Guided Neural Program Synthesis"
_ICLR.cc/2019/Conference_

### Official Review · AnonReviewer3 · 2018-11-03
**Nice idea of using execution information for guiding the synthesizer**

**Rating:** 7
**Confidence:** 5

**Review:**

This paper presents two new ideas on leveraging program semantics to improve the current neural program synthesis approaches. The first idea uses execution based semantic information of a partial program to guide the future decoding of the remaining program. The second idea proposes using an ensembling approach to train multiple synthesizers and then select a program based on a majority vote or shortest length criterion. The ideas are evaluated in the context of the Karel synthesis domain, and the evaluation shows a significant improvement of over 13% (from 77% to 90%).

The idea of using program execution information to guide the program decoding process is quite natural and useful. There has been some recent work on using dynamic program execution in improving neural program repair approaches, but using such information for synthesis is highly non-trivial because of unknown programs and when the DSL has complex control-flow constructs such as if conditionals and while loops. This paper presents an elegant approach to handle conditionals and loops by building up custom decoding algorithms for first partially synthesizing the conditionals and then synthesizing appropriate statement bodies.

The idea of using ensembles looks relatively straightforward, but it hasn’t been used much in synthesis approaches. The evaluation shows some interesting characteristics of using different selection criterion such as shortest program or majority choice can have some impact on the final synthesized program.

The evaluation results are quite impressive on the challenging Karel domain. It’s great to see that execution and ensembling ideas lead to practical gains.

There were a few points that weren’t clear in the paper:

1. Are the synthesis models still trained on original input-output examples like Bunel et al. 2018? Or are the models now trained on new dataset comprising of (partial-inputs-->final-output) pairs obtained from the partial execution algorithm?

2. In algorithm 2, the algorithm generates bodies for if and else branches until generating the else and fi tokens respectively. It seems the two bodies are being generated independently of each other using the standard synthesizer \Tau. Is there some additional context information provided to the two synthesis calls in lines 8 and 9 so that they know to produce else and fi tokens?

3. Is there any change to the beam search? One can imagine a more sophisticated beam search with semantic information can help as well (e.g. all partial programs that lead to the same intermediate state can be grouped into 1).

---

> ### Author Response · Authors · 2018-11-20
> **Response and clarification**
>
> Thanks a lot for your encouraging comments! We respond to your questions below:
>
> 1. For training set construction, we built a new training set with partial execution information obtained from the original training set, and to make it clearer, we have included the training set construction approach in Section 5.1, with details in Appendix B.
>
> 2. For your question about handling else and fi tokens, we do not need any special handling of the else and fi tokens in Algorithm 2, except that the dataset is constructed differently. The details are explained in Appendix B. In particular, the true branch (ending with else) and the false branch (ending with fi) use different IO pairs. The synthesizer trained with such a dataset can learn to generate the correct tokens respectively.
>
> 3. For your question about the change to the beam search, no, we are using the same beam search proposed in (Bunel et al. 2018). We agree that a more sophisticated beam search has the potential to further improve the performance, but our main point is that the Exec algorithm can improve over any existing training technique, thus we did not modify the beam search in order to highlight the improvement obtained using the Exec algorithm. We will leave the exploration of different beam search algorithms as an interesting future direction.

---

### Official Review · AnonReviewer2 · 2018-11-03
**Ok paper, could be written more clearly**

**Rating:** 7
**Confidence:** 4

**Review:**

This paper proposes guiding program synthesis with information from partial/incomplete program execution. The idea is that by executing partial programs, synthesizers can obtain the information of the state the (partial) program ended in and can, therefore, condition the next step on that (intermediate) state. The paper also mentions ensembling synthesizers to achieve a higher score, and by doing that it outperforms the current state-of-the-art on the Karel dataset program synthesis task.

In general, I like the idea of guiding synthesis with intermediate executions, and the evaluation in the paper shows this does make sense, and it outperforms the SOTA. The idea is original and the evaluation shows it is significant (enough). However, I have two major concerns with the paper, its presented contribution, and the clarity.

First, I cannot accept ensembling as a contribution to this paper. There is nothing novel about the ensemble proposed, and ensembling, as a standard method that pushes models that extra few percentage points, is present in a lot of other research. I have nothing against achieving SOTA results with it, while at the same time showing that the best performing model outperforms previous SOTA, which this paper orderly does. However, I cannot accept non-novel ensembling as a contribution of the paper.

Second, the clarity of the paper should be substantially improved:
- my main issue is that it is not clear how the Exec algorithm (see next point too) is trained. From what I understand Exec is trained on supervised data via MLE. What is the supervised data here?  Given the generality claims and the formulation in Algorithm 1/2, and possible ways one could use the execution information, as well as the fact that the model should be end-to-end trainable via MLE, it seems to me that the model is trained on prefixes (defined by Algorithm 1/2) of programs. Whether this is correct or not, please provide full details on how one can train Exec without using RL.
- By looking at Table 3, it seems that the generalization boost coming from Exec (I’m ignoring ensembling) is higher enough, and that’s great. However, it’s obvious that the exact match gain by Exec is minute, implying that the proposed algorithm albeit great on the generalization metric, does not improve the exact match at all. Do you have any idea why is that? Is that because Exec is trained via MLE and the Exec algorithm doesn’t add anything new to the training procedure?
- how do algorithm 1 and 2 exactly relate? I guess there is a meaning of ellipses in Lines 1 and 13, however, that is not mentioned anywhere. Is the mixture of algorithm 1 and 2 (and a non-presented algorithm for while loops) the Exec algorithm? How exactly are these algorithms joined, i.e what is the final algorithm?
- while on one side, I find some formalizations (problem definitions, definition 1, semantic rules in table 2) nicely done, I do not see their necessity nor big gains from them. In my opinion, the understanding of the rest of the paper does not depend on them, and they are well-described in the text.
- the paper says that the algorithm “helps boost the performance of different existing training algorithms”, however, it does so only on the Bunel et al model (and the MLE baseline in it), and albeit there’s mention of the generality, it has not been shown on anything other than those two models and the Karel dataset.
- do lines 6-7 in Algorithm 2 recurse? Does the model support arbitrarily nested loops/if statements?
- The claim that the shortest principle is most effective is supported by 2 data points, without any information on the variance of the prediction/dependence on the seed. Did you observe this for #models > 10 too? Up to what number?
- In table 3, is Exec on MLE? Could you please, for completeness, present the results of Exec + RL + ensemble in the table too?
- summarization, point 3 - what are the different modules mentioned here? Exec/RL/ensemble?

Minor issues, remarks, typos:
- table 1 position is very unfortunate
- figure 1 is not self-explanatory - it takes quite a lot of space to explain the network architecture, yet it fails to deliver meaning to parts of it (e.g. what is h_t^x, why is it max-pooled, what is g_t, etc)
- abstract & introduction - “Reducing error rate around 60%” absolute percentage points seem like a better evaluation measure (that the paper does use). Why is the error rate reduction necessary here?
- figure 2 - why is the marker in one of the corners, and not in the cell itself?
- Algorithm 1, step 4, is this here just as initialization, so S is non-empty to start with?
- Table 2 rule names are unclear (e.g. S-Seq-Bot ?)
- Table 3 mentions what Exec indicates twice

---

> ### Author Response · Authors · 2018-11-20
> **Clarification and revision (Part 1/2)**
>
> We highly appreciate your comments and suggestions on improving the presentation of this work! We have incorporated them in our revision, and we respond to your concerns and questions below.
>
> One of your major concern is about our claim on the novelty of our ensemble approaches. It is true that ensemble is a well-accepted approach in machine learning. However, we find that one neglected piece is the use of available input to justify a model’s output, which is not possible for many machine learning tasks such as machine translation and image recognition. For input-output program synthesis, once a program is generated, we can easily verify whether the prediction could be correct by executing it with the given input-output pairs. In this way, we can easily remove invalid programs from the ensemble, which improves the performance. Albeit its simplicity and effectiveness, to the best of our knowledge, this idea has not been applied in any previous work. Thus, we think this idea deserves to be populated to more audience in the neural program synthesis community. In particular, we propose: (1) verifying the predictions and filtering out those that are inconsistent with input-output specification before ensembling; and (2) the Shortest ensemble principle, and we observe that it achieves a better result than the Majority Vote principle in many cases, especially when the number of models in the ensemble is small. Both ideas leverage unique properties of the program synthesis task, and are not explored in existing work.  We also revise our introduction section to further explain why we emphasize our new ensemble approaches.

---

> > ### Author Response · Authors · 2018-11-20
> > **Clarification and revision (Part 2/2)**
> >
> > Second, we have revised our paper to address your clarity concerns. Here is a detailed list of responses:
> >
> > - For how to incorporate our Exec algorithm into existing training techniques, we have included the training set construction approach in Section 5.1 with details in Appendix B. After the dataset is constructed, the neural synthesizer can be trained with both supervised learning and reinforcement learning.
> >
> > - About the minor improvement of the exact match, we have added a detailed explanation of why the Exec algorithm is not designed to optimize the exact-match accuracy, as well as why exact match accuracy is not as important as generalization accuracy for real-world applications. This can be found on page 8, the paragraph starting with “Note that the improvement of Exec on the exact match accuracy is relatively minor”.
> >
> > - To describe our Exec algorithm more precisely, we rewrite Algorithm 1 and 2 in a more formal way, rather than providing illustrative pseudo-code in the previous version. Now, Algorithm 1 (Exec) includes the condition when ExecIf and ExecWhile are called, and Algorithm 2 (ExecIf) illustrates how it calls Exec to generate the branches. The ExecWhile algorithm is deferred to the appendix.
> >
> > - About the necessity of our formalizations, we provide the formalization to make the discussion precise, and remove as much confusion as possible. For example, in the previous version, the illustrative style presentation of Algorithm 1 and 2 makes you confused about their precise design. This is also the reason why we turn them into more formal ones. We believe the formalization helps clarify potential confusion, so we leave them as is. We are also happy to move part of them to the appendix if it is preferred.
> >
> > - By “helps boost the performance of different existing training algorithms”, we mainly indicate supervised learning and reinforcement learning algorithms that we have evaluated in our paper rather than different models. We have revised our claim in Section 3.2 to make the statement more precise. Also, we reorganize Table 3 to illustrate more clear about what we mean by training algorithms.
> >
> > - For your question “Do lines 6-7 in Algorithm 2 recurse?”,  yes, they do. See line 7 and 10 in Algorithm 1 as well as line 5-6 in Algorithm 2 for the recursive calls. In our evaluated dataset, the programs have a recursion level of up to 5.
> >
> > - For your questions about ensembling more than 10 models, we extend our evaluation to include up to 15 models in each ensemble, which improves the best performance a little bit. As demonstrated in Figure 3, the majority vote principle always achieves a slightly better generalization accuracy than the shortest principle when at least 9 Exec + RL models are included in the ensemble. Meanwhile, the single model accuracy in the ensemble does not vary much; for example, for Exec + RL models, the mean and standard deviation of a single model accuracy are 85.70% and 0.36% for generalization, and 39.32% and 0.25% for exact match.  For random seed selection, we are using the standard pseudo-random number generator (PRNG) in PyTorch, and we didn’t see a clear correlation between the model performance and the random seed selection.
> >
> > - For your questions about Table 3, we have reorganized Table 3 and included more results for completeness.
> >
> > - For your questions about “different modules”, we indicate our two proposed approaches: Exec and Ensemble. We have revised the bullet to be “The different modules of our proposed approaches, i.e., execution-guided synthesis and ensemble techniques, ...” to make this point more precise.
> >
> > - We have revised our paper to address your minor comments.

---

> > > ### Comment · AnonReviewer2 · 2018-11-25
> > > **Thank you for addressing the issues**
> > >
> > > Seeing your reply and the revisions made in the paper, I am more than happy to increase my score.
> > >
> > > Edited: Just to clarify it, the largest weight of my score was on the issue of the ensemble contribution. Seeing how you clearly outlined the added bonus of being able to verify the correctness of the synthesised program to improve ensembling, you made the issue go away, hence the score increase.

---

> > > > ### Author Response · Authors · 2018-11-25
> > > > **Thank you for the update!**
> > > >
> > > > Thanks a lot for your update! We are very glad to see that our revision addressed your concern, and thank you again for your constructive suggestions on improving our paper!

---

> > > > ### Comment · Area_Chair1 · 2018-11-30
> > > > **Thank you for engaging in the discussion and reconsidering your assessment**
> > > >
> > > > This is an important part of ICLR's review system, and of the scientific process as a whole, so your engagement is noted and appreciated.

---

### Official Review · AnonReviewer1 · 2018-11-05
**Forward search planning**

**Rating:** 7
**Confidence:** 2

**Review:**

The authors introduce two techniques:

One is (old school) forward search planning https://en.wikipedia.org/wiki/State_space_planning#Forward_Search
for input/output-provided sequential neural program synthesis on imperative Domain Specific Languages with an available partial program interpreter (aka transition function)(from which intermediate internal states can be extracted, e.g. assembly, Python).
Previous work did:
  which_instruction, next_neural_state = neural_network(encoding(input_output_pairs), neural_state)
This technique:
  which_instruction = neural_network(encoding(current_execution_state_output_pairs))
  next_execution_state = vectorized_transition_function(current_execution_state, which_instruction)

The second one is ensembles of program synthesizers (only ensembled at test-time).


Guiding program synthesis by intermediate execution states is novel, gets good results and can be applied to popular human programming languages like Python.

Pros
+ Using intermediate execution states
Cons
- State space planning could be done in a learnt tree search fashion, like e.g. Monte Carlo Tree Search
- Ensembling synthesizers at test time only
- why not have stochastic program synthesizers, see them as a generative model, and evaluate top-k generalization?

Page 7
Table 3 line 3: "exeuction" -> "execution"

---

> ### Author Response · Authors · 2018-11-20
> **The alternative approaches are not as effective as the beam search**
>
> Thank you for the encouraging comments, and new ideas to evaluate!
>
> First, for MCTS, we consider it as yet another training approach in addition to supervised learning (SL) and reinforcement learning (RL), which is orthogonal to our main contribution of the Exec algorithm. In fact, Exec algorithm is designed to be combined with any training algorithm that can effectively train the underlying synthesizer. By evaluating SL and RL, we think we have demonstrated this point. In addition, we have reorganized Table 3 to list existing training algorithms as a separate column to make it clearer that our technique can be applied in different training setups.
>
> On the other hand, MCTS is effective especially when the ground truth label (or score) is hard to compute for a state (like in the Go game). In our problem, however, we can easily verify whether a generated program satisfies the input-output specification or not. Therefore, a beam search is sufficient to achieve a high accuracy at test time. Also, MCTS typically requires more computation than the beam search approach for inference. Thus, we prefer a beam search algorithm for program synthesis.
>
> Second, about your comment on ensemble, yes, we only use ensemble at test time. We are unclear about a good way of applying ensemble during training time in our setting, and we would appreciate it if the reviewer can provide more details.
>
> Last but not least, we can easily adapt our model to be stochastic. In particular, the synthesizer could randomly sample program tokens from the softmax output probability distribution, rather than always pick the top-scored tokens as in the beam search. However, in doing so, we found that most generated programs are incorrect. Using our single Exec + RL model with the best performance, we repeat the following experiment 64 times, where 64 is the beam size in our evaluation: for each sample in the testset, we run the stochastic synthesizer described above, and evaluate the overall accuracy. The mean accuracies among all runs are 19.86% (exact match) and 45.15% (generalization), with standard deviations of 0.48% and 0.96% respectively.
>
> We further evaluate the top-64 accuracy of this stochastic approach in the following way. For each test sample, we keep sampling until either (1) a program that matches the input-output specification is generated; or (2) there have been 64 invalid programs synthesized. In doing so, the accuracies are 39.32% (exact match) and 85.84% (generalization) respectively, which are not better than using the beam search. Therefore, beam search is more effective in our problem.
>
> We found that these additional experimental results are not crucial to support our main contributions, but if the reviewer thinks they deserve to appear in the paper, we are happy to incorporate them as well.

---

> > ### Comment · AnonReviewer1 · 2018-11-26
> > **Thanks for clarification**
> >
> > Thanks for responding how beam search is an efficient and effective way of searching for/generating interesting points in the space of programs.
> >
> > As for training an ensemble, there are at least two ways of doing it in your setup.
> > 1. The "gradient boosting" way (https://en.wikipedia.org/wiki/Gradient_boosting) where you iteratively train a new model to fill-in the gaps of the current ensemble.
> > 2. If you had a differentiable ensemble voting mechanism (e.g. average all the networks predictions), then the whole ensemble model would behave like your current base model (as you would be able to compute its log-likelihood).

---

> > > ### Author Response · Authors · 2018-11-26
> > > **Thanks for the explanation!**
> > >
> > > Thank you for your explanation! Unfortunately, we do not have enough time to implement these ideas and report the results before the end of the rebuttal period. But we believe these techniques are orthogonal to be applied to further improve our main techniques. We will try gradient boosting and include the results in our camera-ready version. Our ensemble approach is not differentiable by itself, but we will consider extending the current ensemble approach for training as future work.

---

### Public Comment · (anonymous) · 2018-11-06
**A minor note**

Dear Authors,

Congrats on the really positive reviews. As AnonReviewer3 pointed out ("the recent work on using dynamic program execution in improving neural program repair"), please consider citing the paper [1] to acknowledge the prior work. Anyway very nice work! Congrats again!

[1] Dynamic Neural Program Embedding for Program Repair

---

> ### Author Response · Authors · 2018-11-20
> **We have cited the paper in our revision**
>
> Thank you for pointing out the related paper! This is a very interesting work, and we have discussed it in our revision (see Section 6).

---

### Author Response · Authors · 2018-11-20
**Response and revision**

We thank all reviewers for the constructive feedbacks! We have revised the paper with the following major changes to incorporate the comments:

- We revise the introduction to better explain why we emphasize our new ensemble technique.

- We add a description in the caption of Figure 1 to better explain existing input-output neural program synthesis architectures.

- We rewrite Algorithm 1 and 2 to make them more formal and eliminate the confusions.

- In Section 5.1, we add explanations about training set construction and training algorithms to make our evaluation setup more precise.

- We reorganize Table 3 to more clearly separate different components of each approach. Meanwhile, we also add more results and explanations in Section 5.2 for completeness.

The paper is inevitably longer after adding more content. Now it is 9 pages, which is more than the recommended page number of 8. We think it is helpful for readers to better understand our paper, but if the reviewers have concerns about its length, we are happy to defer certain materials into the appendix to fit the main body into 8 pages.

---

### Meta-Review · Area_Chair1 · 2018-12-13
**Great work**

**Confidence:** 4
**Recommendation:** Accept (Poster)

**Metareview:**

This paper presents a system which exploits semantic information of partial programs during program synthesis, and ensembling of synthesisers. The idea is general, and admirably simple. The explanation is clear, and the results are impressive. The reviewers, some after significant discussion, agree that this paper makes an import contribution and is one of the stronger papers in the conference. While some possible improvements to the method and experiment were discussed with the reviewers, it seems these are more suitable for future research, and that the paper is clearly publishable in its current form.